# OpenReview forum: "RAFT: Reward rAnked FineTuning for Generative Foundation Model Alignment"
_TMLR — Accepted by TMLR_

### Review · Reviewer_Ubny · 2023-09-02

**Summary Of Contributions:**

The paper proposes a way to finetune large-scale generative models of images or text with top-K supervised learning. K samples are produced from the model, and the model is finetuned to produce the highest reward sample. This works better than simple supervised fine-tuning (SFT), and PPO. The method is evaluated on LLAMA-7B and Stable Diffusion v1.5. There are extensive ablations.

**Audience:**

Yes

**Broader Impact Concerns:**

Broader impact concerns are sufficiently addressed in the intro.

**Claims And Evidence:**

Yes

**Requested Changes:**

I was not able to find information on the computational overhead. How does the method compare to PPO in terms of training wall-clock time, memory utilization, and FLOPs? How does this change with different hyperparameters?

**Strengths And Weaknesses:**

Strengths
- The method is novel and interesting
- The evaluation is comprehensive
- The method's empirical performance is good

Weaknesses
- The computational overhead of the proposed method is unclear
- Only three methods are evaluated: SFT, PPO, and the proposed method

---

> ### Author Response · Authors · 2023-09-08
>
> Thanks for your efforts in reviewing our paper and also your constructive reviews! According to your suggestion, we have conducted experiments to compare RAFT with PPO in terms of the computations and we elaborate our new results as follows.
>
> We conducted experiments for both the RAFT and PPO algorithms without early stopping, and the model is considered to be convergent if it subsequently oscillates around a fixed reward level for four consecutive iterations. We take the wall-clock time as the main metric because the FLOPs may not be a proper criteria in this case where both forward operation and backward operation are involved for LLMs. This is because the parallelizations of forward and backward could be quite different. In other words, A forward operations may take similar FLOPs with B backward operations, but their consuming time can be quite different. Specifically, with some quick math (or we can leverage the calculator from huggingface https://huggingface.co/spaces/hf-accelerate/model-memory-usage), loading a LLaMA-7B with bf16 requires 12.34 GB but training using Adam requires 49.35GB due to the gradient information. Therefore, we can use a much more larger forward batch size than backward operations. Since RAFT and PPO consist of both forward operation and backward operation, the FLOPs may not accurately reflect the real training time.
>
> **Computational result.** We report the wall-clock time of RAFT with sampling temperature $\lambda=1.0$ and different $K$, averaged over three independent runs. For $K \in \{8, 16, 32\}$, the wall-clock times are 5h, 6.05h, and 7.65h, respectively. As $K$ increases, the inference time grows, which is the main reason why a larger $K$ leads to a longer overall training time. On the other hand, we note that $K=16$ and $K=32$ typically converge faster with 10-12 iterations, while $K=8$ takes about 15-18 iterations to converge. The faster convergence rate partially compensates for the extra inference cost and helps to mitigate the overhead associated with loading models when RAFT switches between different stages of RAFT training. In comparison, the fastest-performing PPO configuration, with a KL penalty of 0.1 and LoRA training, converged in approximately 8.7h (PPO with a KL penalty of 0.01 converges with approximately 13.5 hours), which is slower than all the RAFT experiments with full training. We remark that we report the training time for one round of experiments with well-tuned parameter. Based on our experience, in practice, the complicated hyper-parameter configuration of PPO may also require more efforts in parameter search. For stable diffusion, since back-propagation cannot be done in the diffusion process, PPO cannot support the alignment of diffusion models. Thus, we mainly discuss the computation overhead in this part. We found that the wall-clock time of backward gradient update is 13.9x than forward. Moreover, the time consumption score computation and ranking is almost negligible (less than 10%). Thus, the computation overhead of RAFT algorithm is economic compared with conventional fine-tuning. Our RAFT algorithm does not even need the real samples to improve the diffusion model.
>
>
> **Memory utilization.** For RAFT, the memory utilization typically does not change with different hyper-parameters. We can describe RAFT in a more specific way as follows:
>
> - We load the generative model, sample a batch of prompts, and generate responses for them with the largest possible forward batch size;
> - We load reward model to score the samples in first step and filter the samples with the highest reward;
> - We fine-tune the generative model with the filtered samples.
>
> In particular, we note the bottleneck of memory happens only on the third step, where backward operations are involved. Therefore, as long as we can fine-tune a generative model when the memory resource permits, we can align the generative model with RAFT.
>
> In contrast, as we mentioned in the paper, the PPO algorithm requires loading multiple LLMs at the same time. With the 8xA40 (48G) in hand, we can only run PPO with LoRA training to avoid the out-of-memory error. We have used the largest batch size within the range where the GPU memory permits to achieve the fastest training in our experiments. Considering the diffusion models, the memory consumption of RAFT is the same as plain fine-tuning. Thus, there is not any memory burden for RAFT algorithm.

---

> > ### Comment · Reviewer_Ubny · 2023-10-30
> > **Acknowledged**
> >
> > Thanks for the update! I believe this analysis will improve the paper.

---

### Review · Reviewer_PJbo · 2023-09-30

**Summary Of Contributions:**

The paper proposes a simple but seemingly effective technique to align generative models by using a reward model (trained on human preference data) to then rank the potential training data, discarding the ones that the reward model rejects. Then, the model can be finetuned on the "chosen" data which, since it contains more samples than the reward model would find acceptable, causes the model to be better aligned. However, the benefit of this method is the possibility to go far beyond a small (potentially limited) "human preference" set and the model can be finetuned on a much wider range of content. The paper claims that this might mitigate the "alignment tax" over using RLHF.

**Audience:**

Yes

**Claims And Evidence:**

No

**Requested Changes:**

Please see the weaknesses above. Specifically, I would like to see some key points discussed/answered:

1. Does the reward model generalize? i.e., Will it work reliably in distributions that are different. i.e., the holdout set, while being unseen, is still in the same format, with limited amount of variations (which is the whole reason why we need something like RAFT in the first place). If the model is unreliable in a wide-variety of text, that reduces the applicability of the method.
2. Maybe the reward model does not need to be *VERY* accurate as long as it generates some signal about ranking "good" samples higher than bad ones. Could the paper do some ablations to study to what degree does RAFT rely on the reward model?
3. If the reward model is trained on a different human preference dataset, would the results still hold? (or alternately the automatic eval is done on a different dataset) (Maybe SHP?)
4. I would encourage putting some additional details to the main text so that supplementary truly acts as "additional" materials.

**Strengths And Weaknesses:**

**Strengths:**

**[S1] Good results:** The results measured using various automated metrics show clear improvement compared to RLHF, PPO, or SFT.

**[S2] Potentially a new way to think about model alignment:** The premise of the paper is sound and could be a great new way to think about model alignment.

**[S3] Clarity:** The paper is clearly written, and easy to read, and the experiments and model details are thoroughly explained. I did have to go to the supplementary materials quite a few times to get full details. But, on the whole, the paper is exceptionally clearly written.

**Weaknesses:**

**[W1] Reliance on reward model:** The central weakness of this work is the hypothesis that the reward model will choose good samples to do SFT on, which will then, overall improve results (not just automatic metrics, which is a good proxy but could potentially be gamed). This has not been sufficiently demonstrated to me. The experiment is a bit circular because it is understandable that choosing more (in the wild) samples that look more like the positive training samples and (potentially) even look like the positive examples in the test set, will improve results on the automatic metrics on the test set. This reliance on the reward model takes a few different facets which are sub-weaknesses on their own:

- The paper uses not only the training data but also half of the test data to train the reward model. This indicates the desire to increase the "coverage" of the test set. Could this result in somewhat gaming the automatic eval metric?
- How does the proposed method work as the factor of accuracy on the holdout set?

---

> ### Author Response · Authors · 2023-10-03
>
> Thanks for your efforts in reviewing our paper and also your insightful suggestions! We have revised our paper based on your suggestions.
>
>
> **Q1 Reliance on the reward model**
>
> Your comments get to a fundamental point of RLHF: the imperfection of reward modeling. How to train a better reward model is a central but largely under-explored problem in RLHF literature. We would like to clarify that the reward modeling part in our paper is a standard procedure following Instruct-GPT [1], where the reward model is a maximal likelihood estimator of the theoretical preference model, i.e., the Bradley-Terry model [2]. Our study is based on the implicit assumption that conducting reward optimization wrt the MLE-induced reward model is helpful to our alignment goal. Meanwhile, we highlight that the predominant PPO-based approach is also based on the hypothesis that "the reward model will choose good samples" (and the PPO is more sensitive to the scale of the reward signal), otherwise the alignment process will converge to undesired model.
>
> The main focus of this paper is to present an alternative approach for reward optimization for a **given a proper reward function**. Therefore, improving reward model (such as generalization ability, multi-datasets performance...) may go beyond the scope of this paper.
>
> Nevertheless, we are glad to share some initial attempts on the impact of reward modeling on RAFT which can be an important supplement to the current paper for motivating future studies. We also add more criteria for a thorough model evaluation.
>
> We are happy to share our understandings and results as below.
>
>
> **Q1-continue: involving the hand-out test set in the reward modeing.**
>
> In our experiments, we choose to involve the prompts in the test set only for a more stable and reliable test procedure, instead of deliberately increasing the coverage for the RAFT and PPO algorithms.
>
> We also highlight that the involvement will NOT change our conclusions and our delibrated design is to further make sure that reward model make precise verdict in our experiments.
> In our early stage of our experiments, we does not involve the test set in the reward modeling, and the main results and conclusions are the same. We redo the experiments to get a better reward model to ensure reliable conclusions. Our motivation are illustrated below.
>
>
> We divide the dataset into (1) training set; (2) hand-out test set: first half of HH-RLHF test set; (3) second half of HH-RLHF test set. We use (1) + (2) to train the reward model because the reward model trained by (1) only achieves a test accuracy of 75.79% on (3) but a 94% accuracy on the training set. Therefore, involving the hand-out set into the reward modeling can largely improve the reliability of the test stage.
>
>
> Furthermore, we would like to argue that the accuracy improvement on the hand-out set would not hack the automatic eval metric for either the RAFT or PPO because we only use the reward model to measure the OUTPUTS of the RAFT and PPO, instead of the original responses in the HH-RLHF set, to report the automatic metrics. We have an analogue in human learning: the teacher (the reward model) has seen some similar "questions" of the final exam and can better evaluate the exam "answers". The students (the LLMs to be aligned) are taught by in-class questions (SFT) and get the guidance with their own answers (RAFT/PPO). They can never have access to the hand-out set before the test stage.
>
>
> Moreover, we do agree that the current metrics alone may not exhaustively reflect the capacity of the models. To this end, we add comparisons with the GPT-4 models and human evaluations to enhance our evaluations.
>
> *GPT-4 Evaluations. For each comparison, we did 100 competions. We ask GPT-4-0613 to verdict the goodness of two models. For each pair, we did two experiments to remove the impact of the input order. GPT-4 answers Win (W), Lose (L), Tie (T) for each experiment. For final verdict, we record the average of two experiments: Win (WW, WT, TW), Lose (LL, LT, TL), Tie (WL, LW, TT).*
> |  | Win | Lose | Tie |
> | -------- | -------- | -------- | -------- |
> | RAFT vs PPO     | 65     | 32  |3     |
> | RAFT vs PPO (KL=0.05) |69 |28 | 3 |
> | RAFT vs RAFT (K=8)     | 48      | 37      |15     |
>
>
> *Human Evaluations. We randomly distribute the comparisons to 7 human experts and evaluate each pair manually without seeing the label.*
> |  | Win | Lose | Tie |
> | -------- | -------- | -------- | -------- |
> | RAFT vs PPO     | 66     | 14     |20     |
> | RAFT vs PPO (KL=0.05) |44  |32  | 24|
> | RAFT (K=32) vs RAFT (K=8)     | 40       | 24       |36     |
>
> In general, both GPT-4 and human evaluations suggest that RAFT outperforms PPO significantly. Meanwhile, we can see that with a larger K, the performace of RAFT algorithm can be improved. We observe a concurrence between GPT-4 and human assessments; however, human evaluators tend to be more conservative, resulting in a greater selection of 'Tie' outcomes.

---

> ### Author Response · Authors · 2023-10-03
>
> **Q2 Does the reward model generalize? i.e., Will it work reliably in distributions that are different. i.e., the holdout set, while being unseen, is still in the same format, with limited amount of variations (which is the whole reason why we need something like RAFT in the first place). If the model is unreliable in a wide-variety of text**
>
> We agree that the generalization ability and OOD performance of the RM is critical and fundamental for RLHF. We would like to elaborate on our findings and also our understandings as follows.
>
> *We would like to summarize the first situation as: mild distribution shift with similar alignment goals.*
>
> First, by following the paradigm of Instruct-GPT [1], the test accuracy of the Open-LLaMA-3B-RM is 75.79% and Open-LLaMA-13B-RM is 81.73% on the test set we use in reward modeling. Therefore, for a given dataset (HH-RLHF), with fixed alignment goals (helpful and harmless), the reward model still achieves a satisfactory accuracy on the test set (in-domain generalization). This is also supported by the fact that the model test reward is always positively related to the training reward. Due to resource constraint, we are not able to test larger models and we expect that larger LLMs (70B or 175B) can give even better results.
>
> However, we point out that we are using the RM to evaluate the responses of the aligned model, instead of the original responses in the HH-RLHF dataset, where distribution shift issue is inevitable. In this case, we are carding about the OOD performance of the RM. In general, we believe that the performance of the reward model on the hand-out set is reliable (i.e. the reward model generalizes well), as we have also observed that RAFT-K32 consistently defeats RAFT-K8 in both of the GPT-4 evaluation and Human evaluation.
>
> **Q3 If the reward model is trained on a different human preference dataset, would the results still hold? (or alternately the automatic eval is done on a different dataset) (Maybe SHP?)**
>
> *We would like to summarize the second situation as: multi-dataset reward modeling and different alignment goals.*
>
> If we have multiple preference dataset or multiple goals (might be controversial sometimes), the reward modeling could be challenging. It largely depends on whether the alignment goals are similar. In practice, LLaMA2 [3] and GPT4 [6] tend to split the dataset according to the alignment goals (helpfulness v.s. safety) and combine them with LLM-based classifier to give a more accurate feedback.
>
> For a case study, we conduct an experiment on chatbot arena (24K), Open Assistant (8K), and HH-RLHF (112K). We train single-dataset reward models for each dataset, and then test the reward model on other datasets with 5000 randomly samples and report the accuracies in the following table.
>
> | Dataset         | open assistant | chatbot | hh_rlhf |
> | -------------- | -------------- | ------- | ------- |
> | open assistant | 69.5           | 61.1    | 58.7    |
> | chatbot        | 66.5           | 62.7    | 56.0    |
> | hh_rlhf        | 69.4           | 64.2    | 77.6    |
>
> Even though the HH-RLHF-RM does not train on other two datasets, the HH-RLHF-RM achieves the best accuray for HH-RLHF and Chatbot, and also nearly best accuracy for Open-assistant. This may be because the helpfulness is the common goal of these three datasets. On the other hand, the number of samples could also be an important factor to the generalization across datasets, as both Chatbot and Open-assistant perform worse on HH-RLHF. This may be also because they do not explicitly take the safety into consideration when creating the datasets.
>
> Thus, direct generalization across preference datasets could be challenging because the alignment goals of different preference datasets can be inconsistent and even controversial. But in our case study, the HH-RLHF-RM generalizes well to some datasets with the same helpfulness goal.
>
> **Q3-part1 Maybe the reward model does not need to be VERY accurate as long as it generates some signal about ranking "good" samples higher than bad ones.**
>
> RAFT relies on the ranking provided by the reward model, while PPO further relies on the scale of the signals. So RAFT may be less sensitive to the noise of the reward modeling, which we believe is an advantage of RAFT over the PPO. For instance while the RM attains satisfactory accuracy, it can encounter calibration issues (as shown in the Figure 7 in the revised version of paper), which means that the confidence of our RM, predicted according to the Bradley-Terry model, does not correspond with the actual accuracy of its answers. This implies that the RM might exhibit confident inaccuracies in its predictions.
>
> The calibration issue could largely hurt the performance of the PPO [4], while RAFT is more robust than PPO because it only relies on the ranking signals so it may not need to be that accurate. We presume that this is one of the reasons why RAFT is better in our human/GPT-4 evaluations.

---

> ### Author Response · Authors · 2023-10-03
>
> **Q3-part2 Could the paper do some ablations to study to what degree does RAFT rely on the reward model?**
>
> We follow [5] to study the reward over-optimization issue of RLHF. Specifically, we train GPT2-RM, GPT-NEO-1.3B-RM as the proxy reward models. On the other hand, we view the more powerful Open-LLaMA-3B-RM as an approximate gold RM. Then, we run RAFT with GPT2-RM, GPT-NEO-1.3B-RM but also test the rewards using Open-LLaMA-3B-RM. As expected, we observe the reward over-optimization issue: in the first stage, the gold reward increases as the proxy rewards increase. Then, the model overfit the imperfection of the proxy reward models, so even the proxy rewards continue to increase, the gold reward decreases. See the Appendix A.3 of the revised version for details.
>
>
> **Q4 putting some additional details to the main text**
>
> Thanks for your suggestion. We will try to make the main pages self-contained. To highlight our main contributions, some standard procedures such as the opmization of the reward [1] was left in the Appendix.
>
>
> **References**
>
> [1] Training language models to follow instructions with human feedback.
>
> [2] Rank analysis of incomplete block designs: I. the method of paired comparisons.
>
> [3] Llama 2: Open Foundation and Fine-Tuned Chat Models.
>
> [4] Stabilizing RLHF through Advantage Model and Selective Rehearsal.
>
> [5] Scaling Laws for Reward Model Overoptimization.
>
> [6] GPT-4 Technical Report.
>
>
>
>
> Thanks again for your constructive comments and suggestions. We believe that incoroperating these additional discussions could largely improve the readability and also clarify of our results and paper. Should you have any further questions or concerns, please do not hesitate to bring them up.

---

### Review · Reviewer_SaZp · 2023-10-02

**Summary Of Contributions:**

The paper presents RAFT for finetuning generative models according to a given reward function. RAFT iterates over three steps: (1) draw samples from the generative model; (2) rank the samples based on their rewards; (3) perform supervised finetuning on the top-ranked samples. Experiments show the effectiveness of RAFT for both large language models and diffusion models. Notably, RAFT outperforms PPO for finetuning LLaMA-7B on the HH-RLHF dataset.

**Audience:**

Yes

**Claims And Evidence:**

Yes

**Requested Changes:**

- Clarify the number of reward queries and the wall clock time used for training PPO and RAFT.
- Compare with DDPO, which also provides an experiment for improving aesthetic score. The comparison itself is of interest to the community. It is ok even if DDPO turns out to be better.

**Strengths And Weaknesses:**

Strengths:
- The method is simple, intuitive, and easy to implement, without many hyperparameters to tune. It can serve as a stable baseline for future work on finetuning foundation models.
- Because the data generation and model finetuning is decoupled, the method seems more scalable to larger foundation models and has a lower requirement for computation resources.
- The method can in principle be applied to any generative model.

Weaknesses:
- It seems unclear whether the comparison with PPO (Table 3) is fair. For example, do you use the same number of reward queries? Is there a wall clock time comparison of training with PPO and RAFT?
- There are no baselines in the diffusion model experiments. [DDPO](https://arxiv.org/abs/2305.13301) and [DPOK](https://arxiv.org/abs/2305.16381) are two RL-based approaches for finetuning diffusion models. Would it be possible to compare to DDPO considering its source code has been released?
- The KL weight $\beta$ should also be included in Table 1.

---

> ### Author Response · Authors · 2023-10-07
>
> Thank you for taking the time to review our paper and for your valuable feedback! We have made revisions according to your suggestions.
>
> **Q1 Clarify the number of reward queries and the wall clock time used for training PPO and RAFT.**
>
> Thanks. The computation disucssion is necessary for improving the readability of the paper. Our responses are as follows, and corresponding revisions have been made in the paper.
>
> Considering our experimental setup, both the RAFT and PPO are performed without early stopping, and the model is considered to be convergent if it subsequently oscillates around a fixed reward level for four consecutive iterations. The hyper-parameters of PPO are well-tuned along the way with all our experiences of TRL, and additional hyper-parameter searches were also conducted to achieve its best performance for a fair comparison, with details in Appendix C.
>
> **Computation cost.** We report the wall-clock time of RAFT with sampling temperature $\lambda=1.0$ and different $K$, averaged over three independent runs. For $K \in \{8, 16, 32\}$, the wall-clock times are 5h, 6.05h, and 7.65h, respectively. As $K$ increases, the inference time grows, which is the main reason why a larger $K$ leads to a longer overall training time. On the other hand, we note that $K=16$ and $K=32$ typically converge faster with 10-12 iterations, while $K=8$ takes about 15-18 iterations to converge. The faster convergence rate partially compensates for the extra inference cost and helps to mitigate the overhead associated with loading models when RAFT switches between different stages of RAFT training. In comparison, the fastest-performing PPO configuration, with a KL penalty of 0.1 and LoRA training, converged in approximately 8.7h (PPO with 0.01 KL penalty  converges with about 13.5 hours), which is slower than all the RAFT experiments with full training. We remark that we report the training time for one round of experiments with well-tuned parameter. From our experience, the complicated hyper-parameter configurations of PPO also require more efforts in parameter search. Moreover, we highlight that RAFT trains in an off-policy manner and the inference and policy improvement are decoupled (the policy to improve can be different from the policy to collected samples). Therefore, any techniques to speed up the inference can be readily integrated into the proposed framework. One straightforward option is to leverage the speculative decoding [1] for 2X-3X acceleration in inference with certain LLMs. In contrast, PPO cannot be benefited from these techniques as its backward propagations require the gradient record in the forward pass. For diffusion, since back-propagation cannot be done in the diffusion process, PPO cannot support the alignment of diffusion models. Thus, we mainly discuss the computation overhead in this part. We found that the wall-clock time of backward gradient update is 13.9x than forward. Moreover, the time consumption score computation and ranking is almost negligible (less than 10%). Thus, the computation overhead of RAFT algorithm is economic compared with conventional fine-tuning. Our RAFT algorithm does not even need the real samples to improve the diffusion model. Moreover, our algorithm is much more computational efficient than DDPO (Q2).
>
> **Reward and Model queries.** Since RAFT learns from the best-of-K policy, RAFT does not take advantage in the reward queries. Specifically, RAFT typically uses 5~15X more reward queries than PPO with different choices of K. We note that in the current RLHF framework, RM is also a LLM, and reward query is also part of forward operations in the alignment process. Therefore, more generally, RAFT does not take advantage in terms of forward operations. On the other hand, we note that (1) the PPO structure (actor, critic), and loss function (value loss, probability of the old and current models...) are more complicated, thus PPO need to take multiple forward pass for one sample; (2) RAFT only uses a small subset of the collected samples; (3) RAFT typically converges faster, these factors partially compensate for the overhead in the forward operations of RAFT, and RAFT is preferred in terms of backward propagations. We provide some typical learning curves of both RAFT and PPO in Figure 9 for your reference.
>
> In practice, since the three stages of RAFT are decoupled, we can take full advantage of the GPU and take a much larger batch size. In contrast, PPO loads all the models at the same time, and can only use part of the memory resource at each stage. Meanwhile, the forward operations are cheaper than the backward propagations. For your reference, with 8 x A40 (48G), generating 2048 x 32 samples with LLaMA-7B takes approximately 20 mins in our current implementations, which could be further accelerated with more advanced generation techniques like speculative decoding. These factors together lead to the faster training speed of RAFT in terms of the wall-clock time.

---

> > ### Author Response · Authors · 2023-10-07
> >
> > **Q2 Compare with DDPO, which also provides an experiment for improving aesthetic score. The comparison itself is of interest to the community. It is ok even if DDPO turns out to be better.**
> >
> > Thanks for your suggestion. We have implemented DDPO [2] and try to compare it with our algorithm. We compare both clip score and aesthetic score in our experiments. It is found that we have comparable performance than DDPO in this task. However, we highlight that the computation cost of RAFT is much less than DDPO. One of the reason is that we formulate the problem as a contextual bandit problem for general generative modeling, but the DDPO and DPOK [3] algorithm formulate the iteration of diffusion process as a Markov decision process. The modification makes the algorithm more well-adapted to diffusion algorithms but loses the potential extensions (to other generative models without iterative generation). Meanwhile, the additional computation cost can potentially be a burden. The deliberated design for diffusion process can really inspire as look closer into the structure of some special generative models. Considering the RAFT algorithm, it is possible to compute the reward of intermediate states and perform the best one out of $K$ samples. Both sampling and reward modeling of RAFT might be enriched, which is left as a future work.
> >
> > *Computation Overhead on a single A40*
> > |  | Time/min |
> > | -------- | -------- |
> > | RAFT     |  $8.4$   |
> > | DDPO | $415$ |
> >
> >
> >
> > *In-Domain Comparison*
> > |  | Aesthetic | Clip Score |
> > | -------- | -------- | -------- |
> > | RAFT     |  $6.14_{\pm 0.49}$    | $27.3_{\pm1.4}$  |
> > | DDPO |$6.04_{\pm0.49}$ | $28.8_{\pm1.2}$|
> >
> >
> > *Out-of-Domain Comparison*
> > |  | Aesthetic | Clip Score |
> > | -------- | -------- | -------- |
> > | RAFT     |  $6.07_{\pm 0.60}$    | $26.7_{\pm4.5}$  |
> > | DDPO |$5.76_{\pm0.59}$ |$30.2_{\pm 1.8}$ |
> >
> >
> >
> > **Q3 The KL weight $\beta$ should also be included in Table 1.**
> >
> > Thanks for the suggestion. We have revised the Table 1.
> >
> >
> >
> > [1] Yaniv Leviathan, Matan Kalman, and Yossi Matias. Fast inference from transformers via speculative decoding. In International Conference on Machine Learning, pp. 19274–19286. PMLR, 2023.
> >
> > [2] Kevin Black, Michael Janner, Yilun Du, Ilya Kostrikov, Sergey Levine. Training diffusion models with reinforcement learning. arXiv preprint arXiv:2305.13301 (2023).
> >
> > [3] Ying Fan, Olivia Watkins, Yuqing Du, Hao Liu, Moonkyung Ryu, Craig Boutilier, Pieter Abbeel, Mohammad Ghavamzadeh, Kangwook Lee, Kimin Lee. DPOK: Reinforcement Learning for Fine-tuning Text-to-Image Diffusion Models. arXiv preprint arXiv:2305.16381 (2023).

---

> > > ### Comment · Reviewer_SaZp · 2023-10-16
> > > **Thanks for your response**
> > >
> > > Thank you for your detailed response and additional experiments. I don't have further concerns.

---

### Decision · Action_Editor_xxPz · 2023-11-12

**Recommendation:** Accept as is

**Comment:**

All the reviewers acknowledge the common strengths of the paper, such as 1) good writing quality, 2) strong performance, 3) simple yet novel, and 4) offering a new perspective in RLHF.

Reviewers raised minor comments on the detailed explanations of some experimental designs and computation overhead. The authors addressed the concerns in great detail and stronger experimental results during the discussion. All the reviewers are delighted by the response, and recommended accept.

**Audience:**

There are not many papers on RL fine-tuning large models, and this paper is a good one. The paper shows good insight with extensive validations.

**Claims And Evidence:**

The authors proposed RAFT, which is a method for finetuning generative models using a reward function. It involves generating samples, ranking them by rewards, and finetuning on the top-ranked samples. RAFT outperforms previous methods like PPO in finetuning models such as LLaMA-7B. It aligns models using a reward model based on human preferences, potentially reducing the "alignment tax" of RLHF. RAFT proves more effective than simple supervised fine-tuning and PPO, validated on LLAMA-7B and Stable Diffusion v1.5, with extensive evaluations demonstrating its efficiency.